# Evaluation of Applicability of Minimum Required Compressive Strength for Cold Weather Concreting Based on Winter Meteorological Factors

**DOI:** 10.3390/ma15238490

**Published:** 2022-11-28

**Authors:** Jiahui Cui, Nguyen Duc Van, Feng Zhang, Yukio Hama

**Affiliations:** 1Division of Engineering, Muroran Institute of Technology, Muroran 050-8585, Japan; 2College of Environmental Technology, Muroran Institute of Technology, Muroran 050-8585, Japan; 3Division of Sustainable and Environmental Engineering, Muroran Institute of Technology, Muroran 050-8585, Japan

**Keywords:** minimum required compressive strength, cold weather concreting, early age frost damage, meteorological factors

## Abstract

In this paper, we evaluated the applicability of the minimum required compressive strength for cold weather concreting based on winter meteorological factors. In this study, a compressive strength test, dynamic elastic modulus test, hydration degree test, underwater weighing test, and freeze–thaw test were performed to investigate the effect of compressive strength development at early ages on frost resistance of concrete. In particular, the ASTM equivalent number of cycles ($Cy_{ASTM-sp}$) of various locations was estimated based on winter meteorological factors. The results of experiments showed that the frost resistance of concrete at early ages increases with increased compressive strength. The relative dynamic modulus of elasticity of concrete of 5.0 MPa showed that it can be maintained above 90% within 18 freeze–thaw cycles. In addition, the $Cy_{ASTM-sp}$ results showed that a compressive strength of 5.0 MPa can protect concrete from early age frost damage in all investigated locations, indicating that a compressive strength of 5.0 MPa is the minimum required for safe and reliable cold weather concreting. However, for concrete structures subjected to repeated freeze–thaw cycles, it is necessary to select a higher compressive strength value according to the construction condition.

## 1. Introduction

The most critical factors in cold weather concreting are early age frost damage and the normal development of strength [1,2]. Early age frost damage is a serious problem for concrete structures in cold regions, which is caused by freezing and freeze–thaw cycles during the initial hardening stage. In cold weather concreting, building warm sheds, heating water and aggregates, ensuring sufficient air content in the concrete, and adding antifreeze agents are common methods to protect concrete from early age frost damage [3,4,5,6]. Early age frost damage in concrete can cause serious problems, such as strength reduction, crack generation, an increase in air permeability, and durability deterioration [7,8,9,10]. Hence, it is necessary to ensure that the concrete is sufficiently cured and it reaches a minimum required compressive strength to prevent early age frost damage [11,12,13,14].

The minimum required compressive strength is a reference value defined by experimental results. Previous studies [11,15,16,17,18,19] were conducted to determine the minimum required compressive strength values. Koh [15] proposed that 5.0 MPa should be taken as the minimum required compressive strength value, which was obtained from repeated experiments of freeze–thaw cycles. Powers [11] found that concrete with a compressive strength of 2.9 MPa can resist freezing at an early age. Subsequently, American Concrete Institute Committee 306-66 [16] specified 3.5 MPa as a protective compressive strength value for concrete against a single freeze–thaw cycle. The RILEM guideline [17] for cold weather concreting states that a compressive strength of at least 5.0 MPa can resist one to several freeze–thaw cycles when there is no external water supply with varying water-to-cement ratios, curing temperatures, and cement types. Voellmy [18] posited that if the concrete surface is often saturated with water, a compressive strength of 15.0 MPa is required to resist frost damage. Rastrup [19] also introduced theories and calculations on the minimum required compressive strength. With decades of practice in actual cold weather concreting, it is widely recognized that concrete with the minimum required compressive strength can resist early age frost damage.

However, according to recent studies, even once concrete has reached a compressive strength of 5.0 MPa, freezing and multiple freeze–thaw cycles still have adverse effects on its durability. Koh et al. [20] verified the pertinence of a minimum compressive strength of 5.0 MPa. Concrete specimens were subjected to 30 freeze–thaw cycles between +4 °C and −18 °C in a laboratory test, which is approximately equivalent to freeze–thaw actions that would occur within one year in Korea’s real, natural environment. The frozen concrete was prepared with 28 d 20 °C standard curing. The results showed no difference in compressive strength between frozen concrete and non-frozen concrete. However, frost resistance and chloride ion penetration resistance decreased significantly. Similarly, Choi et al. [21] subjected concrete exhibiting a compressive strength exceeding 5.0 MPa to freezing at −20 °C for 24 h; the compressive strength values of frozen concrete and non-frozen concrete were approximately the same, whereas the frost resistance of frozen concrete was considerably lower than that of non-frozen concrete. Therefore, 5.0 MPa is the minimum required compressive strength to prevent early age frost damage for reasonable strength development. Nevertheless, once concrete is subjected to severe freezing or multiple freeze–thaw cycles at an early age, exceeding the protective capacity of the minimum required compressive strength, the durability of concrete is damaged.

According to the Guide to Cold Weather Concreting of American Concrete Institute Committee 306 [22], the conditions of cold weather concreting occur when the air temperature has fallen to or is expected to fall below 4 °C during the protection period, a period that generally extends from winter to spring. During the period before spring, even if concrete reaches the minimum required compressive strength, its durability may be deteriorated as a result of sudden cold waves, repeated freeze–thaw cycles, and insufficient early age curing according to research findings reported by Koh et al. [20] and Choi et al. [21]. However, there is a lack of related studies on the effect of minimum required compressive strength for cold weather concreting on durability. Hence, it is necessary to investigate the effect of the minimum required compressive strength of concrete on its durability.

Generally, under experimental conditions, evaluation of frost resistance performed well on completely hardened concrete. The test method to determine the frost resistance of concrete is to place concrete into temperature-adjustable equipment, such as the ASTM C 666 [23], JIS A 1148 [24], or GB/T50082-2009 [25], and subject it to 300 freeze–thaw cycles. For example, if concrete with minimum compressive strength is subjected to 300 freeze–thaw cycles based ASTM C 666 [23], it is possible to determine the deterioration behaviors of concrete with the minimum compressive strength within 300 freeze–thaw cycles. Determination of frost resistance of concrete with the minimum compressive strength can help to prevent early age frost damage.

Furthermore, to connect with concrete structures in the natural environment, the frost damage deterioration behavior of concrete at early ages can be estimated by meteorological factors. Considerable progress has already been made in estimating the frost damage deterioration behavior of concrete based on meteorological factors [26,27,28]. Hama et al. [27] proposed an estimation method for the ASTM (American Society for Testing and Materials) equivalent number of cycles ($Cy_{ASTM-sp}$) based on meteorological factors. It can convert the freeze–thaw actions received per year from the temperature and environmental conditions into several freeze–thaw cycles in the standard of ASTM C 666 A [23] method to determine the regional characteristics of frost damage of cement-based materials. It is well known that early age frost damage always occurs after construction from winter to spring. Concrete structures exposed to the natural environment often experience freeze–thaw actions during this period. It is possible to estimate the theoretical number of freeze–thaw cycles of concrete structures experienced in cold regions during this period according to the estimation method of $Cy_{ASTM-sp}$.

As mentioned above, an unclear relationship exists between the minimum required compressive strength and frost resistance at early ages of concrete. Moreover, there is no study using meteorological factors to evaluate the applicability of minimum required compressive strength. Therefore, this study aimed to investigate the effect of compressive strength development at early ages on frost resistance of concrete and perform an objective and correct evaluation of the applicability of minimum compressive strength for cold weather concreting according to meteorological factors.

## 2. Summary of Current Regulation Content about Minimum Required Compressive Strength of Various Countries

In the Recommendation for Practice of Cold Weather Concreting of Architectural Institute of Japan [1], the minimum required compressive strength value for building structure constructions is 5.0 MPa. For civil engineering structure constructions, the minimum required compressive strength range is from 5.0 to 15.0 MPa, which is recorded in the Standard Specifications for Concrete Structures, Materials & Construction of Japan Society of Civil Engineers [29]. Therefore, to grasp the effect of the minimum compressive strength on frost resistance at early ages of concrete, it is necessary to examine not only 5.0 MPa but also larger compressive strength values, such as 10.0 or 15.0 MPa. However, the regulation in the guidelines of various countries is not exactly the same. Hence, it is necessary to summarize the current regulation content of the minimum required compressive strength from different countries.

As shown in Table 1, this study summarized the guidelines for cold weather concreting from Europe, the United States, Canada, Japan, China, South Korea, and Russia. For protecting the concrete from early age frost damage, it can be seen that compressive strength of 5.0 MPa is the most common minimum required compressive strength worldwide, such as in Europe, Japan, China, South Korea, and Russia. Concrete of at least 3.5 MPa will not be damaged by exposure to a single freeze–thaw cycle, according to the American Concrete Institute (ACI). In contrast, the Canadian Standards Association (CSA) recommendation is more conservative and requires that the concrete should reach a minimum compressive strength of 7.0 MPa prior to the first freeze–thaw cycle. Furthermore, from the viewpoint of ensuring long-term durability, the Japan Society of Civil Engineers (JSCE), ACI, CSA, and the Ministry of Land, Infrastructure and Transport (MOLIT) of South Korea suggest that the compressive strength values should be larger than 5.0 MPa. JSCE and MOLIT have the same regulations of minimum compressive strength. For the concrete that exposure to severe weather conditions (multiple freeze–thaw cycles) while the surface is likely to be saturated with water, according to the section size of the thin concrete member, ordinary concrete member, and thick concrete member, the compressive strength values are 10.0, 12.0 and 15.0 MPa, respectively. ACI proposes that the compressive strength should be more than 24.5 MPa when exposed to repeated freeze–thaw cycles while critically saturated. CSA requires that the exterior concrete flatwork needs to attain at least 32.0 MPa when exposed to freeze–thaw cycles and de-icer salts conditions, especially for road constructions in cold regions. In the case of wet concrete continuously subjected to multiple freeze–thaw cycles, larger compressive strength values are required to ensure the long-term durability of concrete. Therefore, the minimum required compressive strength values depend on factors such as weather conditions, size of concrete members, and exposure conditions.

From the summary, it can be established that there is a positive correlation between compressive strength and frost resistance of concrete. However, the effect of compressive strength development at early ages on the frost resistance of concrete is not clear. It is also still unknown that concrete can against how many freeze–thaw cycles with the minimum required compressive strength value. Therefore, the main workflows in this study are, firstly, summarizing the current regulation content of the minimum required compressive strength for cold weather concreting in different guidelines of various countries; secondly, conducting experiments to investigate the effect of compressive strength development at early ages on frost resistance of concrete based on the summary of different guidelines; thirdly, collecting the winter meteorological factors data in a few cold regions from various countries and estimating their $Cy_{ASTM-sp}$; finally, comparing the results of experiments and estimations, and evaluating the applicability of minimum required compressive strength for cold weather concreting.

## 3. Experimental Program and $Cy_{ASTM-sp}$ Estimation Plan

### 3.1. Experimental Materials and Mix Proportions

Concrete specimens in this study were manufactured by ordinary Portland cement (OPC), land sand, coarse aggregate, and tap water. The density of ordinary Portland cement was 3.16 g/cm^3^. The density and water absorption of fine aggregate were 2.73 g/cm^3^ and 1.72%, respectively. For the coarse aggregate, the density and water absorption were 2.68 g/cm^3^ and 1.78%, respectively. The concrete mix proportions are listed in Table 2. The water-to-cement (w/c) ratio was 0.5. The sand-to-aggregate (s/a) ratio was 47.1%. An air-entraining (AE) water-reducing agent (Master Pozzolith No. 70) was used to control air content in fresh concrete, and the target air content was 4.5 ± 1.5%. An AE water-reducing agent is an admixture used to improve workability and resistance to freezing and thawing by entraining a large number of independent fine air bubbles in concrete and reducing its unit water content. It has both the air entrainment action and the cement dispersing action of a water-reducing agent.

### 3.2. Pre-Experiment and Main Experiments

#### 3.2.1. Pre-Experiment

It is necessary to conduct the pre-experiment first before carrying out the main experiments in order to grasp the curing time of different values of the minimum required compressive strength. Table 3 shows the experimental design of the pre-experiment. Compressive strength values of 5.0, 12.0, 18.0, and 25.0 MPa were selected in strength development based on the summary of Section 2. Pre-experiment contained the curing time estimation based on maturity and the compressive strength verification. Besides, the concrete mix proportions of the pre-experiment were the same as the main experiments. The concrete was mixed and poured into cylindrical plastic molds with dimensions of ∅ 100 × 200 mm. All molds were placed in a room at 20 °C and 60% relative humidity for sealing curing based on the estimated time.

The concept of concrete maturity indicates that the combination of curing time and temperature produces a specific strength for a given concrete mixture. The maturity method is used to develop relationships between time-temperature history and concrete compressive strength. Saul [35] originally defined the concrete maturity concept with consideration of the relationship between time, temperature, and strength gain. Furthermore, Koh [36] proposed an estimation method of concrete strength development based on concrete maturity. Hence, the strength development of concrete was estimated in pre-experiment through maturity to approximately calculate the curing time. According to the content of the relationship between strength development and maturity in Recommendation for Practice of Cold Weather Concreting [1], the following Equations (1)–(5) were referred to:(1)$$M=\sum_{0}^{t}\left(T_{c}+10\right)\Delta t\;$$
(2)$$F=F_{\infty }\times exp\left(a\times M_{C}{}^{b}\right)\;$$
(3)F∞=F20 28expa×M20 28b×1−CfT24−20 
(4)$$M_{c}=\left(\sum_{0}^{t}\left(T_{c}+10\right)\Delta t\right)+C_{M}\left(T_{24}-20\right)\;$$
(5)a=a1×F20 28+a2,   b=b1×F20 28+b2 
where M is the maturity factor, (Degree Day, the following is called °D·D); $T_{c}$ is the average concrete temperature (°C) during Δt period (days); F is the compressive strength of concrete (MPa); $F_{\infty }$ is the final strength of concrete (MPa); F20 28 is the compressive strength at 28 days of standard 20 °C underwater curing (MPa); M20 28  is the maturity factor at 28 days of standard 20 °C underwater curing (°D·D); $M_{c}$ is the maturity factor with temperature correction of concrete (°D·D); $T_{24}$ is the average temperature of concrete for 24 h after placing (°C); $a_{1}$, $a_{2}$, $b_{1}$, $b_{2}$, $C_{f}$, and $C_{M}$ are coefficients that depend on the cement types, and the coefficients of OPC were shown in Table 4 according to Recommendation for Practice of Cold Weather Concreting [1].

After estimating the curing time of each specimen and verifying the actual compressive strength using the JIS A 1108 [37], the results of all specimens curing times in pre-experiment were determined.

#### 3.2.2. Main Experiments

Table 5 shows the experimental design of the main experiments. The symbol of the N specimen means the non-frozen concrete as a reference specimen for Series 1 and 2. F and S represent the frozen concrete and required compressive strength value. For example, the symbols F-S5, F-S12, F-S18, and F-S25 mean frozen concrete with 5.0, 12.0, 18.0, and 25.0 MPa, respectively. Especially the specimen of F-T12 in Series 1 was the frozen concrete with a pre-curing time of 12 h. The pre-curing time before freezing of each specimen was determined by the pre-experiment. Experimental items in the main experiments were compressive strength, hydration degree, total porosity, and freezing and thawing resistance, as shown in Table 5. In this study, the main experiments were divided into two series due to different investigation objectives. Figure 1 shows the flowchart of the main experiments. The concrete mixture was manufactured, and then it was cast into ∅ 100 × 200 mm cylindrical plastic molds and 75 × 75 × 400 mm steel molds. Afterwards, the specimens were laid in a room at 20 °C and 60% relative humidity for pre-curing under sealing condition. For the reference specimen, after curing for 28 days, the freeze–thaw test was carried out until 300 cycles according to the JIS A 1148 A method [24].

In Series 1, it contained the case of early age frost damage of concrete (F-T12) and the conditions of concrete with different compressive strength values (F-S5, F-S12, F-S18, and F-S25), which is aimed at evaluating the physical recovery performance of concrete that was subjected to several freeze–thaw cycles at early ages. Based on the pre-experiment results, the compressive strength values of the frozen specimens in Series 1 were determined by the compressive strength test after 20 °C sealing curing for 12 h, 24 h, 44 h, 50 h, and 75 h, respectively. After confirming the compressive strength, the specimens of Series 1 were transferred to an adjustable temperature chamber for freeze–thaw cycles in air condition for 3 cycles, and one cycle was 12 h −20 °C and 12 h +5 °C, which was used to simulate the daytime and nighttime temperatures in the Hokkaido of Japan during winter. After 3 freeze–thaw cycles, the specimens were returned to a room at 20 °C and 60% relative humidity to perform the recovery curing until 31 days. Experimental items were performed, and the freeze–thaw test was conducted until 300 cycles in accordance with the JIS A 1148 A method.

Series 2 was designed to investigate the effect of different compressive strength values on frost resistance of concrete at early ages and assess the deterioration behaviors in 300 freeze–thaw cycles. Before the freeze–thaw test, the 20 °C sealing curing ages of specimens were 24 h, 44 h, 50 h, and 75 h, respectively. Measurement items were carried out after the pre-curing, then the specimens were subjected to 300 freeze–thaw cycles according to JIS A 1148 A method.

### 3.3. Test Methods

#### 3.3.1. Fresh Properties

The slump, air content, and temperature of fresh concrete were measured according to the JIS A 1101 [38], JIS A 1128 [39], and JIS A 1156 [40], respectively.

#### 3.3.2. Hydration Degree Test

After the specified curing time, the samples were cut into 5 mm cube samples from the central area of the specimens. Acetone was used to stop the hydration process of samples for 1 day, and then the samples were dried using the F-drying method [41]. The sample was placed in a vacuum container for water absorption that exceeded 3 h and then put into a drying furnace at 105 °C for 24 h. After mass measurement, the sample was dried in a high-temperature furnace at 1050 °C for 1.5 h. After cooling, the weight of the sample was measured. The hydration degree α was calculated by the following Equations (6) and (7):(6)$$Bound\;water\;content=\frac{m_{1}-m_{2}}{m_{2}}$$
(7)$$\alpha \;\left(\%\right)=\frac{Bound\;water\;content}{0.23}$$
where α is the hydration degree (%); $m_{1}$ is the sample mass after drying at 105 °C (g); $m_{2}$ is the sample mass after drying at 1050 °C (g); 0.23 in the Equation (7) is the mass of water consumed when 1 g of cement is completely hydrated (g).

#### 3.3.3. Underwater Weighing Test

The underwater weighing test was conducted to determine the total porosity of the concrete. Samples were used as a whole specimen without cutting. After boiling the samples in hot water for 6 h, the fire was turned off, and the water was gradually cooled at room temperature. The samples were continued to be kept in the container for 24 h. Subsequently, samples were removed from the container, and the surface moisture was wiped so as to measure the mass of a saturated surface-dry specimen. The mass of the sample underwater was determined by putting the sample into the equipment for underwater weighing. And then, the sample mass of the absolute dry specimen was measured after drying at 105 °C for 24 h. The following Equations (8)–(10) were used to calculate the total porosity:(8)$$V_{t}\left(\%\right)=\left(1-\frac{\rho_{b}}{\rho_{tr}}\right)\times 100\%$$
(9)$$\rho_{b}\;\left(g/cm^{3}\right)=\frac{m_{ad}}{m_{sd}-m_{w}}\times \rho_{w}$$
(10)$$\rho_{tr}\;\left(g/cm^{3}\right)=\frac{m_{ad}}{m_{ad}-m_{w}}\times \rho_{w}$$
where $V_{t}$ is the total porosity (%); $\rho_{b}$ is the bulk density (g/cm^3^); $\rho_{tr}$ is the true density (g/cm^3^); $m_{ad}$ is the sample mass of the absolute dry specimen (g) after drying at 105 °C; $m_{sd}$ is the sample mass of saturated surface-dry specimen (g); $m_{w}=m_{t}-m_{ew}$ is the mass of sample underwater (g); $m_{t}$ is the mass of the equipment and sample underwater (g); $m_{ew}$ is the mass of the equipment underwater (g); $\rho_{w}$ is the density of water (g/cm^3^).

#### 3.3.4. Compressive Strength Test

Compressive strength test was conducted using a ∅ 100 × 200 mm cylinder specimen in accordance with JIS A 1108 [37].

#### 3.3.5. Freeze–Thaw Test

The freeze–thaw test was determined according to the JIS A 1148 A method [24] on specimens with dimensions of 75 × 75 × 400 mm. One freeze–thaw cycle contains 4 h, which is a freezing condition of 2.5 h −18 °C and a thawing condition of 1.5 h +5 °C. The mass loss change and the fundamental transverse frequency of the specimens were measured within 300 cycles. The relative dynamic modulus of elasticity (RDM) and the durability factor (DF) were used to evaluate the resistance of concrete to freezing and thawing. The mass loss change, RDM, and DF of the specimens were calculated as follows:(11)$$W\;\left(\%\right)=\frac{w_{0}-w_{n}}{w_{0}}\times 100\%$$
(12)$$P_{n}\left(\%\right)=\left(\frac{f_{n}^{2}}{f_{0}^{2}}\right)\times 100\%$$
(13)$$DF=\frac{P\times N}{M}$$
where W is the mass loss change (%); $w_{n}$ is the specimen mass at n cycles (g); $w_{0}$ is the specimen mass at 0 cycles (g); $P_{n}$ is the relative dynamic modulus of elasticity (%); $f_{n}$ is the fundamental transverse frequency at n cycles (Hz); $f_{0}$ is the fundamental transverse frequency at 0 cycles (Hz); DF is the durability factor of concrete; P is the relative dynamic modulus of elasticity at n cycles; N is the number of cycles at which P reaches 60% or the number of cycles is 300, whichever is less; M is 300 cycles number.

### 3.4. $Cy_{ASTM-sp}$ Estimation Based on Meteorological Factors

#### 3.4.1. Selection of Meteorological Data of Cold Regions and Period

In this study, several locations in cold regions from different countries were selected for the investigation to comprehensively evaluate the applicability of the minimum required compressive strength for cold weather concreting. The selected locations and meteorological data types are given in Table 6. The meteorological data used in the calculation were collected from the Integrated Surface Dataset (ISD) of the National Oceanic and Atmospheric Administration (NOAA) [42]. The five-year meteorological data from 2015 to 2019 were selected to obtain the average values.

#### 3.4.2. $Cy_{\begin{array}{c} ASTM-sp \end{array}}$ Estimation Method

It is well known that $Cy_{\begin{array}{c} ASTM-sp \end{array}}$ can estimate the freeze–thaw actions that concrete structures have received per year from the natural environment based on meteorological factors. The following Equations (14)–(16) were used to calculate the $Cy_{ASTM-sp}$ results:(14)$$Cy_{\begin{array}{c} ASTM-sp \end{array}}=C\times F\times s\times p\times R_{a90}$$
(15)$$T=-ta\;\min\left(1-Df/Dw\right)$$
(16)$$R_{a90}=4.2T-5.4$$
where $Cy_{\begin{array}{c} ASTM-sp \end{array}}$ is the ASTM equivalent number of cycles with consideration of environmental coefficient (cycles per year); C is the curing conditions; F is the freeze–thaw conditions; s is the insolation conditions; p is the deterioration process coefficient; $R_{a90}$ is the ASTM equivalent number of cycles based on the air temperature (cycles per year), it also means the deterioration process at a degree where the relative dynamic modulus of elasticity is at 100% to 90%; T is the regional coefficient; ta min is the annual extreme value of daily minimum temperature (°C); Df is the number of days for freezing, the number of days that the daily temperature continues to be below 0 °C; Dw is the total number of days for freezing and thawing, the number of days that the minimum temperature of the day is below −1 °C and the maximum temperature of the day exceeds 0 °C.

In addition, because the winter meteorological data were mainly used in this study, the calculation conditions were considered as insolation condition of the northern side (s), air curing condition (C), air freezing-water thawing condition (F), and deterioration process coefficient of 100% to 90% (p). Thus, according to the study content of Hama et al. [27], these environmental coefficient values were determined. that is, s, C, F, and p were 1.00, 0.66, 0.21, and 1.00, respectively.

## 4. Results and Discussions

### 4.1. Pre-Experiment

Figure 2 shows the results of determining the actual compressive strength and curing time. The results showed that all types of specimens achieved the expected compressive strength. Accordingly, the results of the curing time determined in the pre-experiment can be used to set the curing time for each type of specimen in the main experiments. The curing time of S5 ($T_{S5}$), $T_{S12}$, $T_{S18}$, and $T_{S25}$ were 24 h, 44 h, 50 h, and 75 h, respectively.

### 4.2. Main Experiments

#### 4.2.1. Fresh Concrete

Due to the limitations of the experimental conditions, the concrete specimens of Series 1 and 2 were manufactured in two batches. Table 7 presents the results of the slump, air content, and temperature of fresh concrete. It can be found that the fresh concrete in Series 1 and 2 obtained the target range of the slump and air content. The air content was kept at approximately 4.5% by using the AE water-reducing agent. In addition, fresh concrete from Series 1 and 2 exhibited good workability.

#### 4.2.2. Compressive Strength

It is necessary to determine whether the specimens attain the expected compressive strength values before performing the test items. The compressive strength test was measured for F-S5, F-S12, F-S18, and F-S25 in Series 1 and 2. Figure 3 shows the determination result of the compressive strength for each type of specimen. It can be seen that the compressive strength values for each type of specimen in Series 1 and Series 2 met the experimental design requirements considering the dispersion.

Figure 4 shows compressive strength after recovery curing in Series 1. The compressive strength of all types of frozen (F) concrete specimens was roughly the same as that of non-frozen (N) concrete specimens. Especially, the compressive strength of F-T12 was only 1.1 MPa before freezing, which was not met the requirement of the minimum required compressive strength. However, after recovery curing, the strength development was not slowed down. The compressive strength also reached the same degree as that of N. It can be considered that F-T12 did not suffer from early age frost damage. The strength development of F-T12 was not affected by freezing because the AE water-reducing agent was added to the concrete, which ensures sufficient air content and effectively improves the early freezing resistance of concrete. This result is consistent with the findings of Yamashita [43], who reported that the strength development of concrete with an AE agent at early ages is not affected by freezing after the final setting because the air content produced a significant effect.

#### 4.2.3. Hydration Degree and Total Porosity

Figure 5 shows the results of the hydration degree and the total porosity in Series 1 at 31 days. Even if the compressive strength values of F specimens were approximately at the same extent as N after the recovery curing, the hydration degree results of F specimens were lower than that of N, and the total porosity results of F specimens were greater than that of N. It was found that if the pre-curing time of concrete is short, the hydration degree tends to be low, the total porosity tends to increase, and the effect of early age freezing remains. This result is consistent with the previous study [44], which revealed that early age frost damage could reduce the hydration degree of pure cement paste. In addition, although there is a difference in the hydration degree and the total porosity between F specimens and N specimens after the recovery curing, the difference is a little bit, which indicates that recovery curing enabled frozen concrete to achieve a comparable pore structure to that of non-frozen concrete.

The results of the hydration degree and the total porosity in Series 2 before the freeze–thaw test are shown in Figure 6. From F-S5 to F-S25, the hydration degree was increased gradually, and the total porosity was decreased step by step. It showed an increased tendency in the hydration degree and a decreased tendency in the total porosity with the increase in curing time. However, although F-S25 had a compressive strength of more than 25.0 MPa, it still had much difference compared to N on the hydration degree and the total porosity. It illustrated that the hydration reaction of cement was not carried out completely at 75 h.

#### 4.2.4. Frost Resistance of Concrete

Figure 7 shows the results of RDM and weight loss after 300 freeze–thaw cycles in Series 1. As can be seen in Figure 7, the results of RDM and mass change between F specimens and N were almost the same variation tendency until 300 cycles. Therefore, the durability factor results of all specimens in Series 1 were also at the same level, as shown in Figure 8.

According to Figure 4 and Figure 5, in terms of compressive strength, hydration degree, and total porosity, the results of F specimens were near that of N, which demonstrated that the recovery curing could repair the internal deterioration caused by freeze–thaw cycles at early ages. In addition, according to the results of Table 7, the air content of fresh concrete was 4.3% in Series 1. Meanwhile, Hu et al. [45] reported that concrete suffered from freezing after the final setting time, which would lose little service performance of concrete. The dense matrix of hydration products was formed to resist the dilation pressure caused by the ice lens. Generally, when using ordinary Portland cement, the final setting of fresh concrete is completed around 8 h, considering different w/b ratios [43,45]. In Series 1, the pre-curing time periods of F specimens before exposure to early age freezing were 12 h, 24 h, 44 h, 50 h, and 75 h, respectively. The pre-curing time periods of all F specimens were obviously more than 8 h. As discussed above, the recovery curing, the freezing after the final setting, and sufficient air content in concrete may be the main reasons why all F specimens have the same level of resistance to freezing and thawing as that of N.

The results of the RDM and weight loss in Series 2 are shown in Figure 9. It can be seen in Figure 9, the RDM of F specimens decreased more rapidly compared to the reference specimen. The concrete with lower compressive strength showed early frost damage with the increasing number of freeze–thaw cycles.

According to Figure 6, different compressive strength values of concrete represented different denseness degrees of the pore structure inside concrete at early ages. When subjected to repeated freeze–thaw cycles, the aqueous solution in the capillaries and air voids produces dilation pressure due to freezing. Micro-cracks were caused when the strength of the skeleton structure was exceeded [46]. As the number of freeze–thaw cycles increases, the micro-cracks inside concrete continually develop. Thus, in Figure 9a, the greater the compressive strength of concrete, the higher the corresponding hydration degree, the denser the matrix of hydration products, and the more freeze–thaw cycles it can withstand, and vice versa. The variation tendency in weight loss within 300 freeze–thaw cycles, as shown in Figure 9b, is similar to that of the RDM results. With the increase in compressive strength, the number of freeze–thaw cycles improves, and the weight loss gains.

Figure 10 shows the relationship between the compressive strength and durability factor. It can be found that the durability factor increases with increasing compressive strength. The value of R^2^ was greater than 0.9, which implied that there is a high positive correlation between compressive strength and durability factor. Besides, the compressive strength of N can also represent that of all F specimens in Series 1 because they had almost the same degree of compressive strength as that of N. Hence, it was demonstrated that the frost resistance of concrete at early ages depends on the pre-curing time and the hydration degree.

From the above results, it was confirmed that air-entrained concrete could withstand several freeze–thaw cycles at early ages and prevent early age frost damage by ensuring a compressive strength of 5.0 MPa. However, because the hydration reaction process of cement, which is during the compressive strength development at early ages, was not sufficiently advanced, the frost resistance of concrete at early ages was low. On the other hand, the freeze–thaw action from the natural environment in the cold weather concreting period or winter cannot cause damage equivalent to the 300 freeze–thaw cycles in JIS A 1148 A test method. Concrete structures after construction in winter would be expected to get a recovery curing from spring to autumn.

Besides, the numbers of freeze–thaw cycles that the RDM remained above 90% for F specimens in Series 2 were determined that F-S5 at 18 freeze–thaw cycles, F-S12 at 55 cycles, F-S18 at 90 cycles, and F-S25 at 124 cycles. According to the survey results of Section 2 in this paper, the compressive strength of 5 MPa is used as the minimum required compressive strength for cold weather concreting in many countries. Therefore, 18 freeze–thaw cycles of F-S5 would be regarded as the reference values of deterioration occurrence to compare with the estimation results of $Cy_{ASTM-sp}$ in this study.

### 4.3. $Cy_{ASTM-sp}$ Estimation Based on Winter Meteorological Factors

Table 8 shows the calculation results of winter meteorological data of various locations from different countries according to the Integrated Surface Dataset (ISD) of NOAA [42].

$R_{a90}$ and $Cy_{ASTM-sp}$ results of various locations were estimated by Equations (14)–(16), as shown in Figure 11. It was found that $R_{a90}$ is much larger than $Cy_{ASTM-sp}$ in the same location if only considering the air temperature. However, when multiplying by the environmental coefficient, $Cy_{ASTM-sp}$ results of all locations were lower than 18 freeze–thaw cycles a lot. It was illustrated that the minimum required compressive strength of 5.0 MPa has sufficient frost resistance to protect concrete from early age frost damage during the cold weather concreting periods and winter. Therefore, according to the results of Figure 11, it verified that the compressive strength of 5.0 MPa as the minimum required compressive strength for cold weather concreting is practicable. Before exposure to the freeze–thaw cycles, if the concrete is cured to more than 5.0 MPa, the quality of the concrete may be guaranteed. As for some concrete constructions under critical water-saturated environments, such as dams and bridges, it is necessary to select a compressive strength value greater than 5.0 MPa to prevent damage due to freeze–thaw cycles, depending on the construction requirements.

In this study, the applicability of 5.0 MPa as the minimum required compressive strength for cold weather concreting was determined for the first time with the combination of experiments and estimation based on meteorological factors. Moreover, the findings of this study were persuasive by investigating guidelines for cold weather concreting in different countries and winter meteorological data from various locations in the world.

## 5. Conclusions

In this paper, the effect of compressive strength development at early ages on the frost resistance of concrete was investigated, and the $Cy_{ASTM-sp}$ results of various locations were estimated based on winter meteorological factors. The following conclusions can be drawn from the results:Air-entrained concrete that reaches a compressive strength of 5.0 MPa can withstand several freeze–thaw cycles and effectively prevent early age frost damage.Recovery curing significantly improves the compressive strength and frost resistance of early frozen concrete. It can make frozen concrete attain normal strength development and achieve the same degree of frost resistance as non-frozen concrete.For early age concrete subjected to repeated freeze–thaw cycles in a critical water-saturated environment, the frost resistance increases with the increase in compressive strength. For OPC concrete with w/c of 0.5, the numbers of freeze–thaw cycles of the concrete with compressive strength values of 5.0, 12.0, 18.0, and 25.0 MPa that can maintain the relative dynamic modulus of elasticity above 90% were about 18, 55, 90, and 124 cycles, respectively.Using winter meteorological factors from various locations to estimate $Cy_{ASTM-sp}$ shows that 5.0 MPa as the minimum required compressive strength for cold weather concreting is sufficient to protect concrete from early age frost damage. For concrete constructions under critical water-saturated environments, from the viewpoint of concrete quality management, it is necessary to use a compressive strength value greater than 5.0 MPa.

## Figures and Tables

**Figure 1 materials-15-08490-f001:**
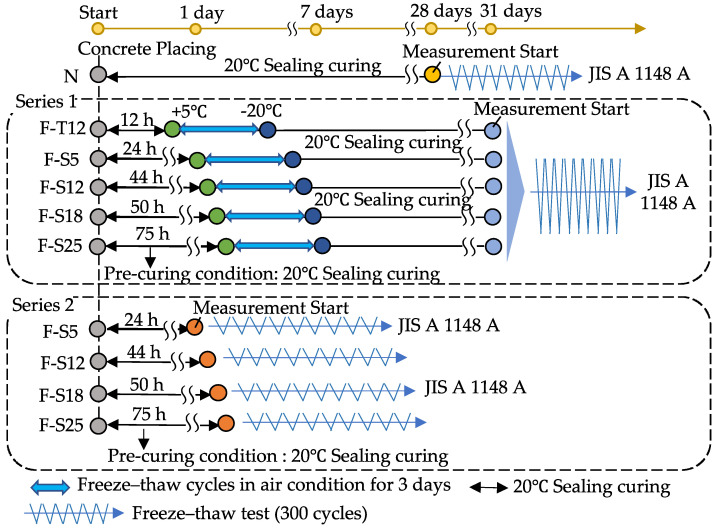
Flowchart of the main experiments.

**Figure 2 materials-15-08490-f002:**
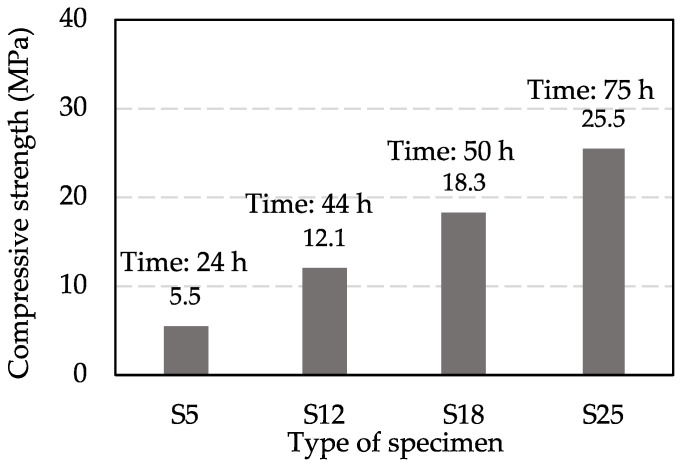
Determination of actual compressive strength and curing time.

**Figure 3 materials-15-08490-f003:**
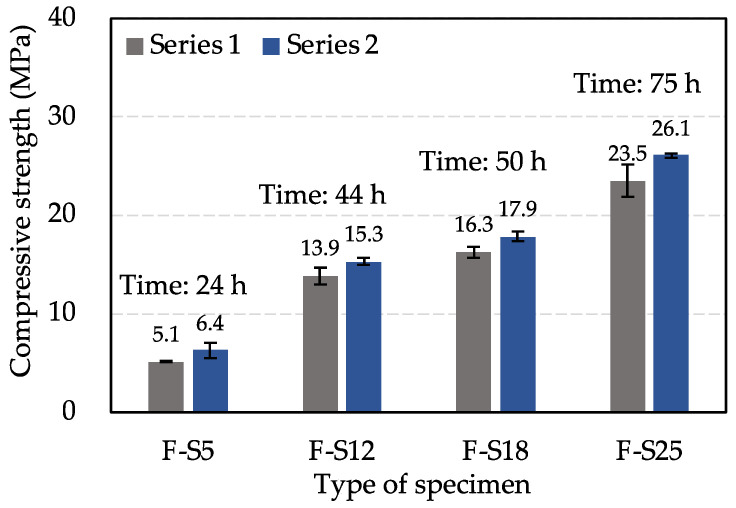
Determination of compressive strength after specified curing time.

**Figure 4 materials-15-08490-f004:**
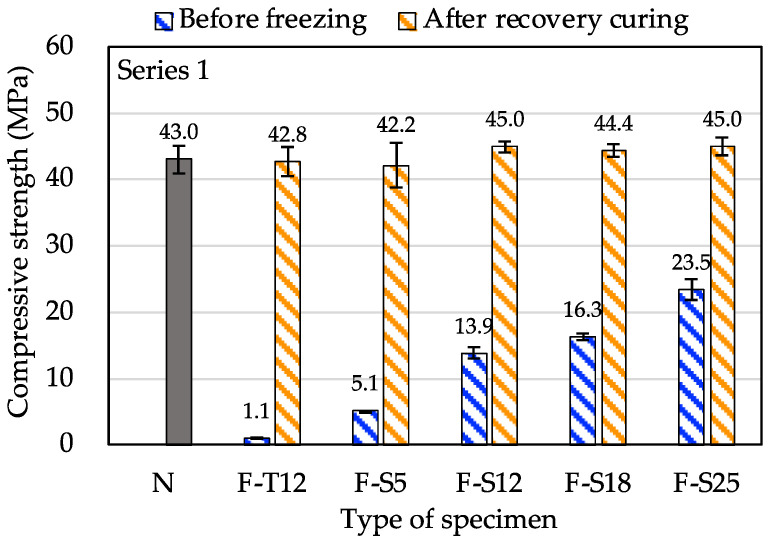
The compressive strength after recovery curing in Series 1.

**Figure 5 materials-15-08490-f005:**
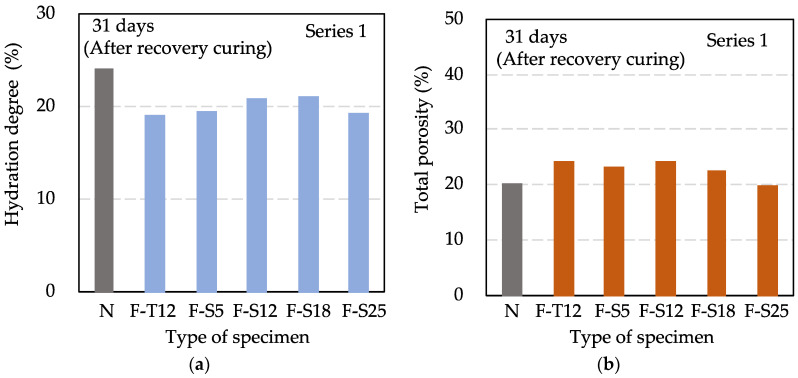
Results of hydration degree (**a**) and total porosity (**b**) in Series 1 at 31 days.

**Figure 6 materials-15-08490-f006:**
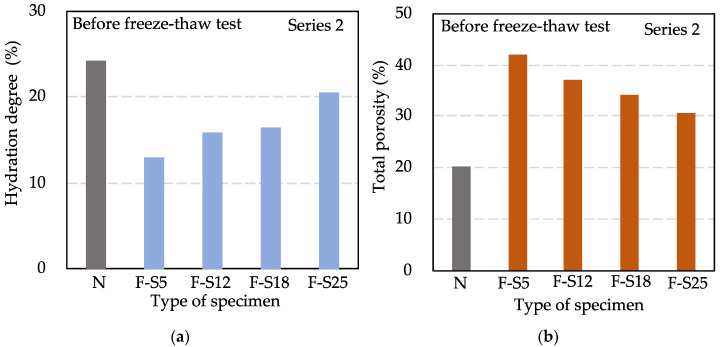
Results of hydration degree (**a**) and total porosity (**b**) in Series 2 before freeze–thaw test.

**Figure 7 materials-15-08490-f007:**
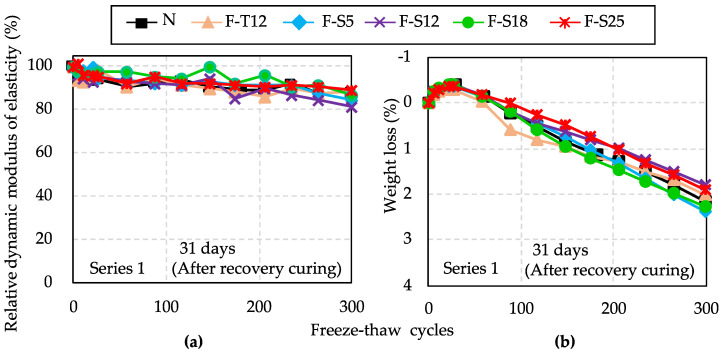
Results of relative dynamic modulus of elasticity (**a**) and weight loss (**b**) after 300 freeze–thaw cycles in Series 1.

**Figure 8 materials-15-08490-f008:**
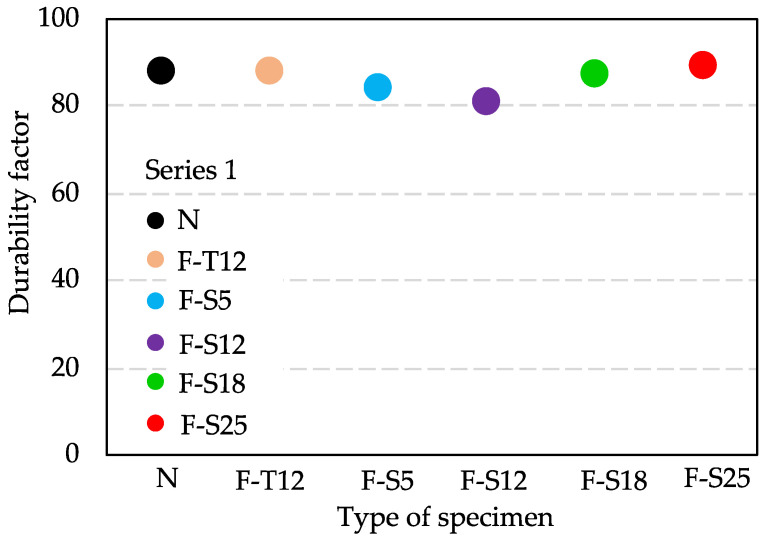
Durability factor results of Series 1.

**Figure 9 materials-15-08490-f009:**
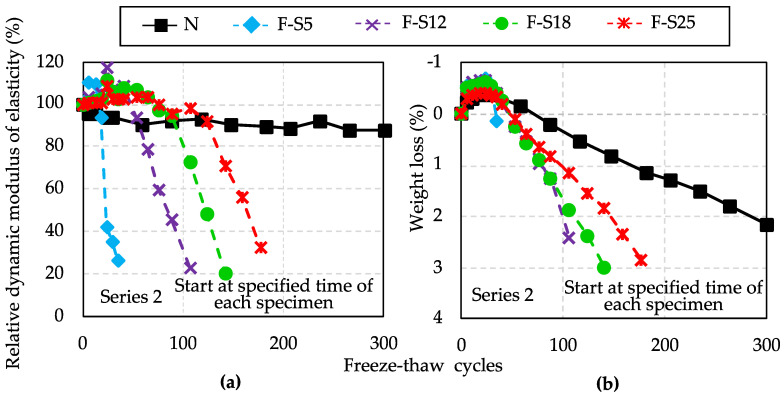
Results of relative dynamic modulus of elasticity (**a**) and weight loss (**b**) after 300 freeze–thaw cycles in Series 2.

**Figure 10 materials-15-08490-f010:**
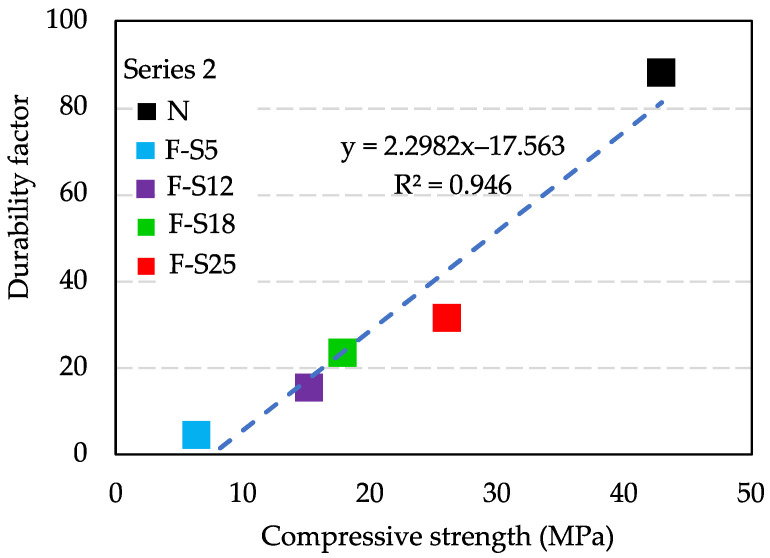
Relationship between compressive strength and durability factor.

**Figure 11 materials-15-08490-f011:**
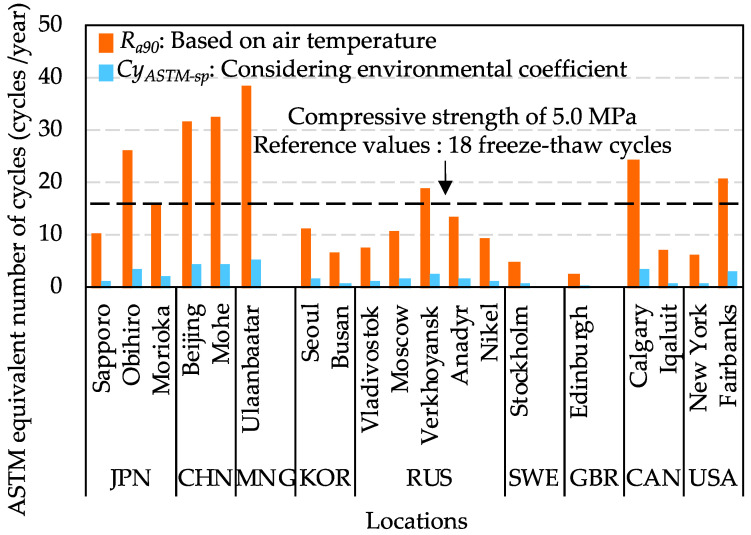
$R_{a90}$ and $Cy_{\begin{array}{c} ASTM-sp \end{array}}$results of various locations.

**Table 1 materials-15-08490-t001:** Regulation content of minimum required compressive strength of various countries.

Guides	Compressive Strength Values
RILEM recommendations for concreting in cold weather [30] (VTT Technical Research Centre of Finland, 1988)	5.0 MPa
Guide to Cold Weather Concreting [22] (American Concrete Institute Committee 306, 2016)	(1) 3.5 MPa (2) When concrete exposed to repeated freeze–thaw cycles while critically saturated: more than 24.5 MPa
Concrete materials and methods of concrete construction/Test methods and standard practices for concrete [31] (Canadian Standards Association, 2014)	(1) 7.0 MPa (2) When exterior concrete flatwork exposed to freeze–thaw cycles and de-icer salts: at least 32.0 MPa
Recommendation for Practice of Cold Weather Concreting [1] (Architectural Institute of Japan, 2010)	5.0 MPa
Standard Specifications for Concrete Structures, Materials & Construction [29] (Japan Society of Civil Engineers, 2017)	(1) When the concrete surface is frequently saturated with water: depending on the *size of the cross-section Thin: 15.0 MPa, Ordinary: 12.0 MPa, Thick: 10.0 MPa (2) When the concrete surface is rarely saturated with water: depending on the size of the section Thin: 5.0 MPa, Ordinary: 5.0 MPa, Thick: 5.0 MPa
Specification for Winter Construction of Building Engineering [32] (Ministry of Housing and Urban-Rural Development of the People’s Republic of China, 2011)	Minimum air temperature ≥ −15 °C, 4.0 MPa Minimum air temperature ≥ −30 °C, 5.0 MPa
Recommendations for the Production of Concrete Work in Winter [33] (Non-profit partnership self-regulatory organization union of construction companies of Ural and Siberia, 2015)	5.0 MPa
Concrete Standard Specification [34] (Ministry of Land, Infrastructure and Transport of South Korea, 2016)	The regulation content is the same as Japan Society of Civil Engineers

* denotes the size of the section: Thin 20 to 30 cm, Ordinary 30 to 90 cm, Thick 90 to 100 cm

**Table 2 materials-15-08490-t002:** Concrete mix proportions.

w/c	s/a (%)	Slump (cm)	Air Content (%)	Unit Weight (kg/m^3^)				AE Water-Reducing Agent (mL/C = 100 kg)
				W	C	S	G	
0.5	47.1	18 ± 2.0	4.5 ± 1.5	175	350	852	957	250

Note: W: water; C: cement; S: sand (fine aggregate); G: coarse aggregate.

**Table 3 materials-15-08490-t003:** Experimental design of pre-experiment.

Symbol	Strength Development (MPa)	Curing Condition	Experimental Items
S5	5.0	Sealing curing	Maturity Compressive strength
S12	12.0		
S18	18.0		
S25	25.0		

**Table 4 materials-15-08490-t004:** Coefficients of Ordinary Portland cement.

Cement Type	$a_{1}$	$a_{2}$	$b_{1}$	$b_{2}$	$C_{f}$	$C_{M}$
OPC	526.9	−37.8	13.34	−1.06	−0.0005	0.680

**Table 5 materials-15-08490-t005:** Experimental design of the main experiments.

Series	Symbol	Pre-Curing Time before Freezing (h)	Strength Development (MPa)	Recovery Curing Conditions	Experimental Items
-	N	-	-	-	Compressive strength, Hydration degree, Total porosity, Freezing and thawing resistance
Series 1	F-T12	12	-	With recovery curing	
	F-S5	24	5.0		
	F-S12	44	12.0		
	F-S18	50	18.0		
	F-S25	75	25.0		
Series 2	F-S5	24	5.0	Without recovery curing	
	F-S12	44	12.0		
	F-S18	50	18.0		
	F-S25	75	25.0		

Note: N: non-frozen concrete; F: frozen concrete; S: required compressive strength value; T: time.

**Table 6 materials-15-08490-t006:** Meteorological data collection program.

Countries	Locations	Period (year)	Data Types
Japan	Sapporo	2015–2019	Annual extreme value of daily minimum temperature, Number of days for freezing, Total number of days for freezing and thawing
	Obihiro		
	Morioka		
China	Beijing		
	Mohe		
Mongolia	Ulaanbaatar		
South Korea	Seoul		
	Busan		
Russia	Vladivostok		
	Moscow		
	Verkhoyansk		
	Anadyr		
	Nikel		
Sweden	Stockholm		
England	Edinburgh		
Canada	Calgary		
	Iqaluit		
America	New York		
	Fairbanks		

**Table 7 materials-15-08490-t007:** Test results of fresh concrete.

Series	Slump (cm)	Air Content (%)	Temperature (°C)
Series 1	18.5	4.3	18.5
Series 2	17.5	4.0	19.0

**Table 8 materials-15-08490-t008:** Calculation results of winter meteorological data of various locations.

Countries	Locations	Data Types		
		Annual Extreme Value of Daily Minimum Temperature (ta min, °C)	Number of Days for Freezing (Df, day)	Total Number of Days for Freezing and Thawing (Dw, day)
Japan	Sapporo	−11.92	43.20	63.00
	Obihiro	−43.08	48.00	89.40
	Morioka	−10.56	7.00	93.80
China	Beijing	−8.80	12.40	95.20
	Mohe	−42.52	143.00	75.00
Mongolia	Ulaanbaatar	−23.20	127.00	89.20
South Korea	Seoul	−14.22	19.80	60.80
	Busan	−8.20	0.60	39.80
Russia	Vladivostok	−21.74	100.20	29.20
	Moscow	−15.40	66.00	40.60
	Verkhoyansk	−55.40	194.60	45.60
	Anadyr	−37.40	164.60	25.00
	Nikel	−30.20	133.60	28.20
Sweden	Stockholm	−12.30	25.20	38.20
England	Edinburgh	−6.06	0.60	23.80
Canada	Calgary	−28.20	55.60	80.00
	Iqaluit	−38.00	202.80	18.00
America	New York	−15.68	13.60	32.00
	Fairbanks	−34.24	133.20	58.80

## Data Availability

Not applicable.

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
