# Peer review of "Evaluation of Applicability of Minimum Required Compressive Strength for Cold Weather Concreting Based on Winter Meteorological Factors"

_materials, 2022, doi:10.3390/ma15238490_

Round 1

Reviewer 1 Report

Why dynamic elastic modulus is not mentioned in the abstract?

Line 31-32: Consider changing the sentence “The most critical things in cold weather concreting are preventing early age frost damage and guaranteeing a normal strength Development” with “The most critical things in cold weather concreting are early frost damage and the normal development of strength” or something like that.

Line 34: “cyclic freeze-thaw cycles” ??

Line 35: Consider changing “securing” with “ensuring”.

Line 46: “Powers [11] reported that the hydration process consumes the water in fresh concrete during strength development...” This statement is pretty obvious. “... and also calculated the degree of dryness at which frost damage will not occur.” And what was the result found?

Line 51: once freezing?

Please, revise the English, there are several typos over the manuscript.

Line 56: “Rastrup [19] also introduced the theories and calculations on the minimum required compressive strength.” And what they discovered?

Line 80: Incomplete sentence.

What type of cement was used?

Table 5 should include additional information to differentiate series 1 from series 2, not just in the text. Otherwise, the Table is unnecessary. Also, the nomenclature does not allow a quick assimilation. Maybe F-T24-S5, F-T44-S12 or something like that.

Section 2 has information repeated in section 1. I don't see the need to separate it into two sections with the same line of thinking. Some sentences by the way are quite similar, see line 49 and 140.

Line: “It is also still unclear different minimum required compressive strength values can exactly resist how many freeze-thaw cycles.” Unclear sentence.

Line 254 – “were replaced” / “were placed”

Please, include the meaning of AE (water-reducing agent).

Figures 3 and 4 – I recommend inserting the error margin.

Line 376 – “was not suffered” / “did not suffer”

Line 404 – diffidence?

Fig. 7a – The scale of figure should be shortened, perhaps between 100 and 75. As shown, it appears that the authors intended to mask the results. I recommend inserting the error margin, also for the other results.

Fig. 9b was not discussed. Why F-S5 did not present continuos weight loss with increasing freeze-thaw cycles as for the other samples.

Line 462 – Consider replacing “securing” with “ensuring”.

Why did authors not investigate samples with compressive strength less than 5 MPa? Maybe 3 MPa would be enough. As stated in Line 49: “American Concrete Institute Committee 306-66 [16] specified 3.5 MPa as a protective compressive strength value for concrete against once freezing.”

The authors have often stated about the minimum compressive strength needed to withstand freeze-thaw strength, but have not investigated or plotted compressive strength against freeze-thaw cycles. Why? This is the most important correlation of the work as it is treated as the main objective. By the way, the authors did not bring a comparison of minimum dynamic modulus of elasticity from the literature.

Round 2

Reviewer 1 Report

Abstract: In this study, compressive strength test, dynamic elastic modulus, hydration degree test, underwater weighing test, and freeze-thaw test were performed to investigate the effect of compressive strength development at early ages on frost resistance of concrete.

The meaning of AE I meant was about the nomenclature, which I assume is “Air-entraining”

Chapters 1 and 2 remain repetitive. I recommend focusing only on the regulations, without citing the same works again. Authors can eliminate the first sentence

“According to the previous studies [11,15–19], it was known that the value of the minimum compressive strength depends on the water-saturated condition of the concrete surface and the numbers of freeze-thaw cycles. Accordingly, due to different uses of concrete structures, there are a few different values for the minimum required compressive strength.”

There are still typos in the manuscript. I recommend using the free tool Grammarly, or other software, to eliminate them.

The authors provide most of the recommendations. The manuscript has sufficient quality (despite experimental shortcomings) for being published.
